# Autoencoders with Intrinsic Dimension Constraints for Learning Low-Dimensional Image Representations

## Abstract

Autoencoders have achieved great success in various computer vision applications. The autoencoder learns appropriate low-dimensional image representations through the self-supervised paradigm, i.e., reconstruction. Existing studies mainly focus on minimizing the pixel-level reconstruction error of an image, while mostly ignoring the preservation of the property that reveals the manifold structure of data, such as Intrinsic Dimension (ID). The learning process of most autoencoders is observed to involve dimensionality compression first, and then dimensionality expansion, which plays a crucial role in acquiring low-dimensional image representations. Motivated by the important role of ID, in this work, we propose a novel deep representation learning approach with autoencoder, which incorporates regularization of the global and local ID constraints into the reconstruction of data representations. This approach not only preserves the global manifold structure of the whole dataset but also maintains the local manifold structure of the feature maps of each point, which makes the learned low-dimensional features more discriminant and improves the performance of the downstream tasks. To the best of our knowledge, existing works are rare and limited in exploiting both global and local ID invariant properties on the regularization of DNNs. Numerical experimental results on benchmark datasets (Extended Yale B, Caltech101 and ImageNet) show that the resulting regularized learning models achieve better discriminative representations for downstream tasks including image classification and clustering.

## 1 Introduction

Deep Neural Networks (DNNs) have been successfully applied to various high-dimensional tasks in machine learning, such as computer vision and natural language processing Chollet (2017); Chen et al. (2018); He et al. (2017). The learning process in DNNs often involves changes in the dimensionality of data representation, e.g., dimensionality expansion and compression Ansuini et al. (2019); Yu et al. (2020); Farrell et al. (2022). This phenomenon is captured through computing the Intrinsic Dimension (ID) of data. ID for a set of data points is the minimum number of variables or parameters needed to describe data points in the ambient space with little information loss Bac et al. (2021). Therefore, ID is a fundamental geometrical property of data representations, especially the property of data with geometric structures on manifold Ansuini et al. (2019); Brahma et al. (2015); Farahmand et al. (2007).

Self-Supervised Learning (SSL) has further unleashed the power of DNNs Doersch et al. (2015); Pathak et al. (2016); Gidaris et al. (2018); He et al. (2022); Bao et al. (2022), where SSL can boost the feature extraction performance for networks on the downstream tasks. AutoEncoder (AE) is a classical and effective SSL paradigm, which gets low-dimensional representations through reconstructing inputs via encoding and decoding operations Bengio et al. (2013). In recent years, amounts of AE variants have been proposed and they incorporate the spatial, temporal, spectral, and context information into pixel-level reconstruction Zhou et al. (2019); Jing & Tian (2020); He et al. (2022). Most AEs involve dimensionality compression first, and then dimensionality expansion, in which ID plays a crucial role. However, to the best of our knowledge, it is still a pending question of how to develop AE that incorporates the ID information into reconstruction loss.

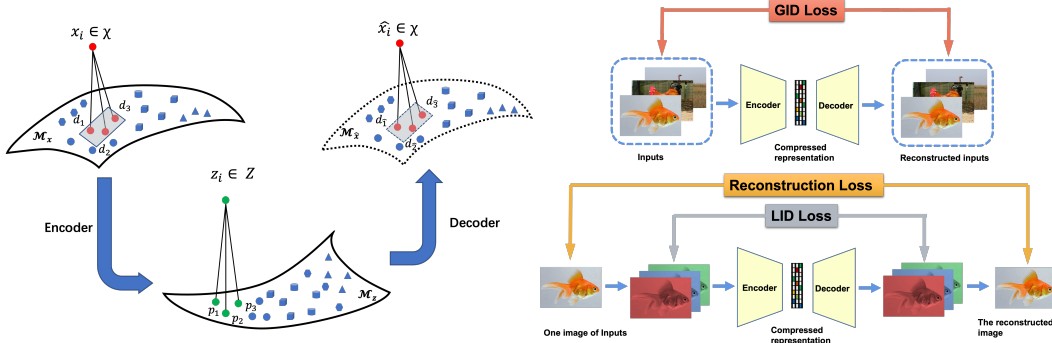

Figure 1: Overview of the training process of proposed AE-IDC. Left: an illustration of the objective of AE-IDC, i.e., getting low-dimensional representations while preserving data manifold structure globally and locally. Right: an illustration of reconstruction loss, GID loss and LID loss, where GID denotes global intrinsic dimension and LID denotes local intrinsic dimension.

In this work, we attempt to answer this question by incorporating both global and local ID information into pixel-level reconstruction to preserve the global and local manifold structure while learning low-dimensional representations, shown in Fig. 1. The main contributions of this paper are summarized as follows:

- The proposed method exploits ID information of data representations from global and local perspectives, i.e., the global ID for whole data points and the local ID for feature maps from each point.

- A generic ID-based regularization is incorporated into the current standard autoencoder framework to develop a novel SSL paradigm, coined as **AutoEncoder with Intrinsic Dimension Constraint (AE-IDC)**, which preserves ID along with reconstruction.

- The proposed AE-IDC achieves more discriminative low-dimensional representations than existing AE models based on Convolutional Neural Networks (CNNs) and Vision Transformers (ViT) Dosovitskiy et al. (2021) on three different scale datasets: Extended Yale B Georghiades et al. (2001), Caltech101 Fei-Fei et al. (2004), and ImageNet Russakovsky et al. (2015).

## 2 RELATED WORK

### 2.1 SELF-SUPERVISED REPRESENTATION LEARNING

AE is a classical SSL paradigm, which consists of two parts: the encoder compresses the input to latent space, and the decoder reconstructs the input. There are various self-supervised learners based on AE, such as Principal Component Analysis (PCA) Goodfellow et al. (2016), Denoising AE (DAE) Vincent et al. (2008), Variational AE (VAE) Kingma & Welling (2014), and Generative Adversarial Networks (GAN) Goodfellow et al. (2014). Leveraging modern network architectures such as Residual Networks (ResNet) He et al. (2016) and ViT, AE can deal with large-scale data efficiently. Recently, Masked AutoEncoders (MAE) He et al. (2022) and its convolutional derivative Gao et al. (2022), driven by the reconstruction from partially random masked inputs, achieve superior performances in downstream computer vision tasks compared to supervised pretraining.

Besides AE-based learners, contrastive learning is another important paradigm in SSL. The contrastive learning methods such as MoCo He et al. (2020), SimCLR Chen et al. (2020) and DINO Caron et al. (2021) do not require the model to be able to reconstruct the original input, but instead learn discriminative representations in the embedding space by maximizing the distance between different points and minimizing the distance between different augmented representations from the same single point.

## 2.2 Intrinsic Dimension and Its Estimation Methods

The concept of ID has gained significant attention as it provides a powerful tool to quantitatively measure the intrinsic geometric structure of data representations. It also serves as a lower bound of the dimension reduction and a measurement of the complexity of the dataset. Accurate ID can help to understand the structure of data representations and guide the design of DNNs in terms of width and depth. The overestimation of ID brings additional computational overhead, whereas the underestimation of ID results in significant information loss.

ID estimation methods can be generally grouped into global or local methods Bac et al. (2021). Global methods consider the whole dataset to provide a single Global Intrinsic Dimension (GID) estimation. Typical algorithms include correlation dimension Grassberger & Procaccia (1983), DANCo Ceruti et al. (2014) and TwoNN Facco et al. (2017). Local methods analyze each data point's neighborhoods separately and provide Local Intrinsic Dimension (LID) estimation for each point in the dataset. Typical algorithms are MADE Farahmand et al. (2007), MLE Levina & Bickel (2004), and Geometry-aware MLE Gomtsyan et al. (2019). Both global and local IDs can be re-purposed: global ID can be estimated by combining local ID estimations, while local ID can be estimated by applying global ID estimation within a local neighborhood. In the proposed AE-IDC framework, the adopted ID estimator belongs to the global method.

## 2.3 The Effects of Intrinsic Dimension on DNNs

The change of ID is related to the change of geometrical properties of DNNs such as distribution, distance, and curvature Ma et al. (2018a); Ansuini et al. (2019), so ID is a quantitative characterization for understanding the learning behavior of DNNs from a geometrical perspective. Huang (2018) constructed an ID-based indicator to understand how low-dimensional representations are developed across layers in simplified neural networks. The work in Nakada & Imaizumi (2020); Latorre et al. (2021); Birdal et al. (2021) analyzed the effect of the ID on the generalization of DNNs. Ansuini et al. (2019) found that the ID profile in trained DNNs follows a hunchback shape, i.e. the ID first increases and then decreases, and the ID of the last hidden layer is crucial to the classifier's performance. Ma et al. (2018b) found the shift of ID is an indication of the start of overfit in the learning process of DNNs on the dataset with noisy labels. Jiang et al. (2021) characterized the smoothness of data manifold by the knowledge of data representation's ID.

Besides the effect on the generalization of DNNs, ID has an effect on the robustness of DNNs. Ansuini et al. (2019) attributes the rise of the ID profile to the redundant features, which are irrelevant to final task predictions. Pope et al. (2021) revealed that high dimensional datasets are more difficult for DNNs to learn, and the dataset with higher ID value is more vulnerable to adversarial perturbations. Moreover, the class with a higher ID value is more vulnerable to attacks compared to other classes in the same dataset. Ma et al. (2018a) found that adversarial attacks can raise the local ID value and train a local ID-based detector to remove adversarial examples from the inputs. However, to the best of our knowledge, there are few studies on explicitly controlling GID and LID of data representations to build a self-supervised representation model.

## 3 Methodology

In this section, we propose a novel self-supervised representation learning framework with differentiable constraints for preserving intrinsic dimension information during reconstruction, to achieve low-dimensional and expressive representations through the encoder function.

**Notation.** Let $\mathbf{X} \in \mathbb{R}^{N \times C \times H \times W}$ be a batch of inputs, where $N, C, H, W$ denote the batch size, the channel number, the height and the width, respectively. Let $\widehat{\mathbf{X}}$ be the reconstruction of $\mathbf{X}$. $\widetilde{\mathbf{X}} \in \mathbb{R}^{N \times m}$ is reshaped from $\mathbf{X}$, where $m = C \times H \times W$ is the number of total features with each feature of $\widetilde{\mathbf{X}}$ being centralized for zero-mean. The covariance matrix of $\mathbf{X}$ is denoted by $\mathbf{C} = \widetilde{\mathbf{X}}^{\top} \widetilde{\mathbf{X}}$. The data representation after a linear transformation is denoted as $\mathbf{Y} = \widetilde{\mathbf{X}} \mathbf{W}^{\top} \in \mathbb{R}^{N \times n}$, where $\mathbf{W}^{\top} \in \mathbb{R}^{m \times n}$ is the transformation matrix.

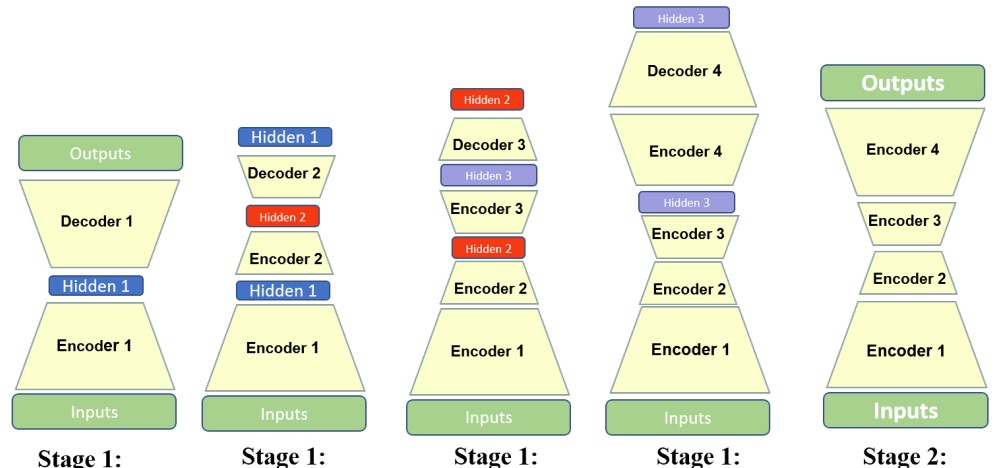

Figure 2: Illustration of Two-stage training for AE-IDC. In this work, a symmetric AE is used. The first half AEs are undercomplete, where the dimensionality of hidden representations is smaller than inputs; while the second half AEs are overcomplete, where the dimensionality of hidden representations is larger than inputs.

## 3.1 CONTROLLING THE GLOBAL AND LOCAL ID OF DATA REPRESENTATIONS.

Neural correlations play vital roles in controlling the dimensionality of neural representations Rajan et al. (2010); Huang (2018). Rajan et al. (2010) introduced an ID estimator from the perspective of the effective number of principal components,

$$\mathbf{ID}(\mathbf{X}) = \frac{(\mathrm{tr}(\mathbf{C}))^2}{\mathrm{tr}(\mathbf{C}^2)} = \frac{(\sum_i e_i)^2}{\sum_i e_i^2} = \left(\sum_i \widetilde{e}_i^2\right)^{-1}, \tag{1}$$

where $\mathbf{C}$ is the covariance matrix of $\mathbf{X}$, and $e_i$ is the eigenvalue of $\mathbf{C}$, and $\widetilde{e}_i = \frac{e_i}{\sum_i e_i}$ is the normalized eigenvalue. This estimator considers the pairwise correlations of inputs and reveals the mechanism underlying how the correlations affect the dimensionality of representations. Inspired by its smoothness with respect to $e_i$ and its differentiability with respect to the network parameters, in this work, we leverage Eq. (1) to estimate the global and local ID of data representations.

We develop a novel self-supervised training framework to compress the original representations into a low-dimensional space for downstream tasks while reducing the variation of the global and local ID of representations along with pixel-level reconstruction. Our framework is an extension of the Stacked Autoencoder (SAE), a hierarchical architecture composed of multiple AEs. In this architecture, the output of one AE serves as the input for the subsequent AE. The training process comprises two stages: initially, the individual AEs are trained in a layerwise manner, followed by global training of all the AEs collectively. Importantly, our framework is not limited solely to SAE; it can also be effectively applied to various other AE variants. The reconstruction loss is the Mean Squared Error (MSE), $L(\mathbf{X}, \widehat{\mathbf{X}}) = \frac{1}{N} \sum_{i=1}^{N} (\widehat{x}_i - x_i)^2$, where $x_i \in \mathbb{R}^{C \times H \times W}$ is a sample in $\mathbf{X}$. By minimizing the reconstruction loss, traditional SAE efficiently learns low-dimensional representations. However, besides reconstruction, this approach ignores the preservation of the property that reveals the manifold structure of data, such as ID.

To address this issue, this paper proposes an effective framework named AutoEncoder with Intrinsic Dimension Constraint (AE-IDC), which introduces two extra constraints (GID and LID) to regularize the learning of AE to preserve global and local information. The GID describes the geometric structure of the subspaces of varying dimensions from points in the batch. To compute the GID, the original inputs (batch, channel, height, weight) are reshaped into a 2D matrix (batch, channel×height×weight). According to Eq. (1), the GID is given by $\mathbf{GID}(\widetilde{\mathbf{X}}) = \frac{\left(\mathrm{tr}(\widetilde{\mathbf{X}}^\top \widetilde{\mathbf{X}})\right)^2}{\mathrm{tr}\left((\widetilde{\mathbf{X}}^\top \widetilde{\mathbf{X}})^2\right)}$.

Considering that feature maps from the same data point are highly correlated, the space of these

---

**Algorithm 1:** Training of AE-IDC.

---

**Input:** $\mathcal{X}$: a dataset of clean examples. $\{AE_j\}_{j=1}^L$: $L$ AEs. $f_{enc}^i$: the encoder of $AE_i$; $f_{dec}^i$: the decoder
       of $AE_i$.
**Output:** The encoder part of the SAE.
\# Layerwise training for each AE.
**for** *i = 1 to l* **do**
    Freeze all the previous AEs $\{AE_j\}_{j=1}^{i-1}$.
    **for** *sample a batch* $\mathbf{X}$ *from* $\mathcal{X}$ **do**
        input of $AE_i$: $\mathbf{X}_i = f_{enc}^{i-1}(\mathbf{X}_{i-1})$, where $\mathbf{X}_1 = \mathbf{X}$.
        compute *loss* according to Eq. (2).
        perform BP to update parameters of $AE_i$ by minimizing *loss*.
\# Global training for whole AEs.
Stack all the encoders $\{f_{enc}^i\}_{i=1}^L$ to construct a SAE.
**for** *sample a batch* $\mathbf{X}$ *from* $\mathcal{X}$ **do**
    compute *loss* according to Eq. (2).
    perform BP to update parameters of SAE by minimizing *loss*.

---

feature maps can be used as the description of the local geometric structure for the point. We regard the ID of such space as the LID of the point. To compute the LID, the original sample (channel, height, weight) is reshaped into (channel, height×weight) first. The term LID is then given by $\mathbf{LID}(x_i) = \frac{\left(\mathrm{tr}(\mathbf{M}^\top \mathbf{M})\right)^2}{\mathrm{tr}((\mathbf{M}^\top \mathbf{M})^2)}$, where $\mathbf{M} \in \mathbb{R}^{C \times HW}$.

It is intuitive and reasonable that the variation of GID and LID between the original inputs and their reconstructions needs to be as small as possible. Motivated by this consideration, we incorporate these two ID constraints into the reconstruction loss function to encourage the SAE to maintain ID. Formally, the overall objective of AE-IDC is

$$\mathcal{J}_{\mathrm{AE-IDC}} = L(\mathbf{X}, \widehat{\mathbf{X}}) + \lambda_1 (\mathrm{GID}(\mathbf{X}) - \mathrm{GID}(\widehat{\mathbf{X}}))^2 + \lambda_2 \sum_{i=1}^N (\mathrm{LID}(x_i) - \mathrm{LID}(\widehat{x}_i))^2, \quad (2)$$

where $\lambda_1$ and $\lambda_2$ are hyper-parameters of the weight of the corresponding regularization. Fig. 5 depicts the performances of AE-IDC with respect to different values of $\lambda_1$ and $\lambda_2$ in the downstream classification task. The pixel-level reconstruction loss is known to be differentiable. The differential of IDC with respect to the weight $W$ of networks can be referred to Appendix A.1. With Eq. (2) being differentiable, it can be incorporated into the backpropagation.

## 3.2 Two-stage Training of Autoencoder with Intrinsic Dimension Constraints

In the framework of AE-IDC, we use an $L-$layer symmetric SAE, where the first $L/2$ layers perform encoding and the second $L/2$ layers perform decoding, shown in Fig. 2. Note that each layer is a separate AE, consisting of encoder and decoder parts. It is noteworthy that the AEs in the first half of the architecture are designed to be undercomplete, whereas the AEs in the latter half are intentionally designed to be overcomplete.

The training framework of AE-IDC is summarized in Algorithm 1. The framework also follows a two-stage paradigm. In the first stage, we train the group of AEs in a layerwise manner. During training the $i^{th}$ AE, we compute the reconstruction, and GID and LID loss according to this AE's input and output, then update the parameters of $AE_i$ locally. In the second stage, we stack all the pretrained encoders and perform end-to-end training to update the parameters of $\{f_{enc}^i\}_{i=1}^L$ globally. At inference, we only take the first half of the trained AE-IDC to perform feature extraction for downstream tasks.

## 3.3 Analysis of Complexity and Convergence

**Complexity Analysis.** We provide a theoretical analysis demonstrating that IDC does not entail significant additional computational overhead. For a single batch, the computational complexity of the $\mathbf{C}$ is $\mathcal{O}(N^2 \times m)$. Then, the computational complexity of $\frac{(\mathrm{tr}(\mathbf{C}))^2}{\mathrm{tr}(\mathbf{C}^2)}$ is $\mathcal{O}(N^3)$. As $N \ll m$, the

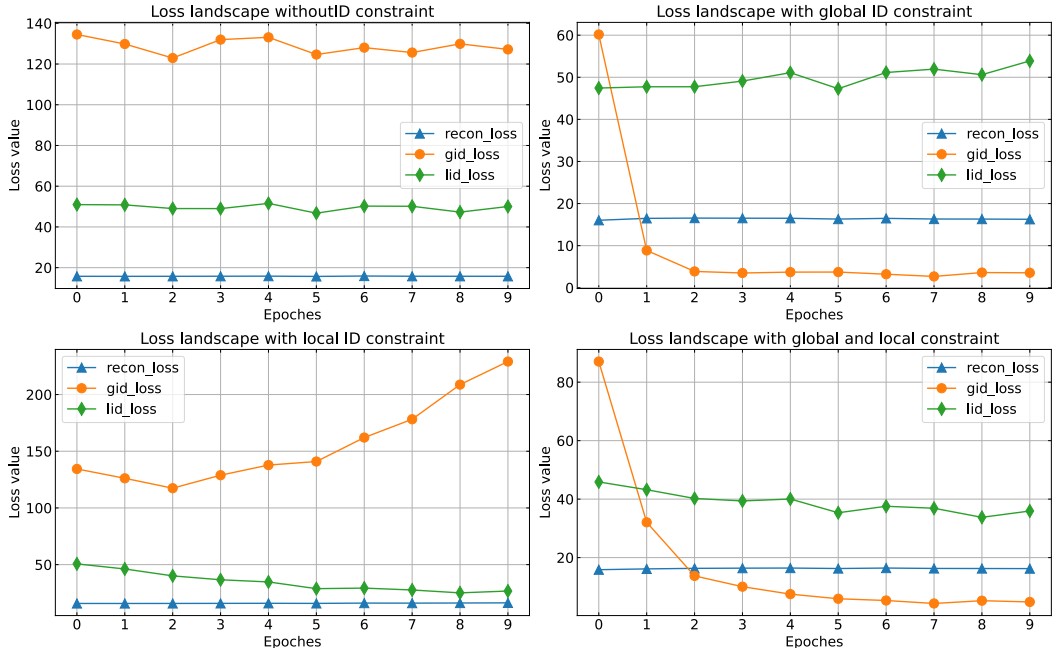

Figure 3: Loss landscapes of MAE and its variants with different ID constraints on ImageNet10. The horizontal axis represents the number of epochs, and the vertical axis represents the loss value. All these four MAE models are initialized with the same pretrained model.

complexity of GID is $\mathcal{O}(N^2 \times m)$, and the complexity of LID is $\mathcal{O}(N^3 \times m)$. Note that, the complexity of MSE is $\mathcal{O}(N \times m)$. All items have the same order of $m$. Therefore, the implementation of IDC does not impose a significant additional burden upon the complexity of the original AE.

**Convergence of the Proposed Constraints.** We empirically evaluate the convergence of the proposed constraints in Fig. 3. From Fig. 3, we can see that GID loss and LID loss quickly converge within ten epochs when models are imposed with corresponding ID constraints.

## 4 EXPERIMENTS

To validate the proposed AE-IDC, we first investigate the effect of IDC during training. Then, we evaluate the feature extraction performance of the proposed AE-IDC on two downstream tasks: image classification and clustering. Finally, we conduct extensive ablation studies to analyze the impact of different components of AE-IDC.

**Datasets.** We evaluate the performance of the proposed method AE-IDC on three widely-used benchmark image datasets including Extended Yale B, Caltech101, and ImageNet. Since running the entire ImageNet is time-consuming, we extract the first ten classes from ImageNet-1K to form a subset, i.e., ImageNet10. ImageNet10 provides a fast test on ImageNet without loss of generality. The configurations about the splitting of datasets in this work are summarized in Table 5 in Appendix A.2.

**Implementation Settings.** We mainly use convolution, deconvolution, maxpooling, and upsampling layers to construct SAE. The details about the architectures of AE used in the following subsections can refer to Appendix A.2. For the large-scale dataset, we adopt the standard ViT-Base (ViT-B) architecture. The weights of regularizers in AE-IDC are set $\lambda_1 = 0.1$ and $\lambda_2 = 0.1$ as default. All experiments in this paper are conducted on an NVIDIA RTX 3090 GPU (2 GB memory). And the codes for the reproduction of our work will be available at Github.

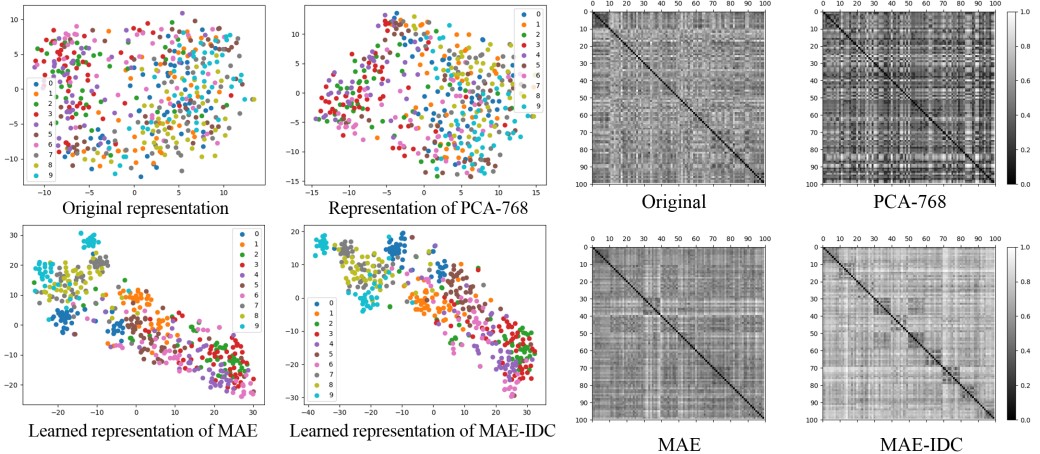

Figure 4: Visualization of self-supervised learned representations on the testing set of ImageNet10 by using $t-$SNE (Left group), and geodesic distance distributions between ten classes from ImageNet10 with each class having ten images (Right group). The geodesic distance is computed approximately by KNN graph Euclidean distance where $K = 15$, and is linearly rescaled to $[0, 1]$ for the convenience of visualization. The distribution of geodesic distance is decentralized in the original space and the linear subspace by PCA. Compared to the original MAE, MAE-IDC is more centralized in diagonal, which indicates it better preserves the manifold structure, i.e., the similarity in the same class and dissimilarity between different classes.

## 4.1 ANALYSIS OF AE-IDC'S LEARNING PROCESS

**CNNs-based AE-IDC.** To simplify the analysis, we analyze the CNNs model used in Extended Yale B, which only consists of four convolutional layers, where the extrinsic dimension of data representations is $(3, 32, 32) - (12, 16, 16) - (24, 8, 8) - (12, 16, 16) - (3, 32, 32)$ from inputs to reconstructions. The figure depicting the whole loss landscapes of two SAE models with and without IDC during two-stage training is attached in Appendix A.3. The reconstruction loss of both models gradually converges at the end of 100 epochs of training in the layerwise and global training stages. But SAE-IDC's reconstruction loss is a bit higher than vanilla SAE. This is because the SAE-IDC is regularized to learn a more abstract embedding feature space instead of simple reconstructions of pixels. For global and local ID loss, SAE-IDC achieves smoother training curves and quicker convergences in the layerwise and global training stages, compared to vanilla SAE. It can obviously observed that for LID loss vanilla SAE does not converge at the first step of layerwise training and the final global training. We suppose that the convergence of reconstruction, global and local ID loss is the reason for SAE-IDC to extract discriminative representations for downstream tasks.

**ViT-based AE-IDC.** MAE He et al. (2022), a form of DAE, is a State-Of-The-Art (SOTA) AE-based self-supervised model. However, MAE only utilizes MSE as its reconstruction target, which ignores the geometric structure information. We will show in the following subsection that the enhanced MAE under the proposed IDC, dubbed MAE-IDC, will unleash the potential of MAE. Fig. 3 shows the loss landscape of four different MAE models. We use the pretrained model from He et al. (2022) to initialize all models. The MAE without ID constraint has reached its optimal point at the start, resulting in its losses remaining nearly constant. Conversely, the MAE variants with ID constraints converge quickly after ten epochs, with the GID loss and LID loss dropping after imposing the corresponding regularizer. All these MAE variants keep the reconstruction ability, shown in Fig. 8 in Appendix A.4.

## 4.2 EVALUATION OF REPRESENTATIONS ON DOWNSTREAM TASKS

We choose image classification and clustering as downstream tasks. The performance of embedded representations is evaluated on the classification task using $K-$Nearest Neighbor (KNN) algorithm, and on the clustering task using $K-$means algorithm. The KNN and $K-$means algorithms provide

Table 1: Classification performance (metric: Average Top-1 Accuracy(%)) on Extended Yale B, Caltech-101, and ImageNet datasets, where a KNN classifier is applied after feature extraction.

| Dataset | Method | Arch. | Dim. | k=5 | k=10 | k=15 |
|---|---|---|---|---|---|---|
| Extended Yale B | SAE | CNNs | (24,8,8) | 82.28 | 81.75 | 79.19 |
| | SAE-IDC | CNNs | (24,8,8) | **86.50** | **83.78** | **81.96** |
| | DAE | CNNs | (24,8,8) | 66.70 | 62.75 | 59.34 |
| | DAE-IDC | CNNs | (24,8,8) | **69.65** | **65.47** | **62.65** |
| | SparseAE | CNNs | (24,8,8) | 82.07 | 81.22 | 81.22 |
| | SparseAE-IDC | CNNs | (24,8,8) | **84.85** | **84.63** | **83.30** |
| Caltech101 | SAE | CNNs | (24,28,28) | 49.49 | 43.39 | 39.66 |
| | SAE-IDC | CNNs | (24,28,28) | **50.33** | **45.48** | **42.20** |
| ImageNet10 | SAE | CNNs | (512,7,7) | 34.94 | 33.47 | 33.26 |
| | SAE-IDC | CNNs | (512,7,7) | **35.77** | **36.19** | **35.56** |
| | MAE He et al. (2022) | ViT-B | 768 | 74.89 | 75.52 | 75.94 |
| | MAE-IDC | ViT-B | 768 | **75.94** | **76.56** | **76.98** |
| | MoCo v3 Chen et al. (2021) | ViT-B | 768 | 27.61 | 28.03 | 29.91 |
| | DINO Caron et al. (2021) | ViT-B | 768 | 73.22 | 73.64 | 76.35 |
| ImageNet-1K | MAE He et al. (2022) | ViT-B | 768 | 49.03 | 45.82 | 43.94 |
| | MAE-IDC | ViT-B | 768 | _49.14_ | _45.93_ | _44.06_ |
| | MoCo v3 Chen et al. (2021) | ViT-B | 768 | 27.61 | 28.03 | 29.91 |
| | DINO Caron et al. (2021) | ViT-B | 768 | 67.32 | 63.97 | 62.30 |

a fast test, without the need to carry on a heavy end-to-end fine-tuning, and also provide relative fairness for comparison. The number of iterations to run $K-$means is ten.

**Results on Classification Tasks.** To demonstrate the generality of the proposed algorithmic framework, we apply this framework to two other widely used AE variants, i.e., DAE and sparse AE. As seen in Table 1, for CNNs-based models, AE-IDC outperforms AE without IDC on all three datasets by $1\% \sim 5\%$. We also compare AE-IDC with SOTA self-supervised learning methods on ImageNet10 and ImageNet-1K. For a fair comparison, we let the compared methods use its public official fine-tuned models without any modification. as shown in Table 1. Though only improving the best performance by $0.1\%$ on ImageNet-1K, the training process of MAE-IDC is more concise and easy to understand.

**Results on Clustering Tasks.** The metrics for evaluation are Adjusted Mutual Index (AMI) and Adjusted Rand Index (ARI). All models are pretrained on ImageNet-1K. The results in Table 2 show MAE-IDC's advantage over MAE on clustering task. This is in coordinated with visualization results in Fig. 4. Although MAE-IDC falls behind the SOTA contrastive learning DINO, the gap shrinks on Caltech101.

Table 2: Clustering performance (metric: ARI∥AMI) of the proposed AE-IDC with comparisons to three SOTA self-supervised methods, where $K-$means algorithm is applied after feature extraction. The configuration of architecture and embedding dimension is the same with Table 1.

| Method | ImageNet10 | Caltech101 |
|---|---|---|
| MAE He et al. (2022) | 0.280∥0.451 | 0.293∥0.532 |
| MAE-IDC | **0.291∥0.467** | **0.313∥0.542** |
| MoCo v3 Chen et al. (2021) | 0.090∥0.189 | 0.047∥0.112 |
| DINO Caron et al. (2021) | 0.491∥0.645 | 0.328∥0.576 |

## 4.3 Ablation studies

In this subsection, we ablate the design of AE-IDC, and analyze the impacts of different elements in the loss function, the stagewise training and the weights of regularizers on the performance of AE-IDC, respectively.

**Elements in Loss Function.** We demonstrate the classification accuracy of models trained with three variants of loss Function: **Reconstruction+GID**, **Reconstruction+LID** and **GID+LID**. As

Table 3: Impact of each component in the proposed loss function (Eq. (2)) to the KNN recognition rate on the Extended Yale B.

| Loss Item | KNN Acc |
|---|---|
| Reconstruction (Baseline) | 82.28 |
| Reconstruction + GID | 83.99 |
| Reconstruction + LID | 83.78 |
| Reconstruction + GID + LID | **86.50** |
| GID + LID | 59.98 |

Table 4: Impact of each stage in the proposed two-stage training paradigm of AE-IDC in Algorithm 1 to the KNN recognition rate on the Extended Yale B.

| Training Paradigm | KNN Acc |
|---|---|
| Baseline | 82.28 |
| Layerwise training | 83.88 |
| Global training | 85.92 |
| Layerwise + Global training | **86.50** |

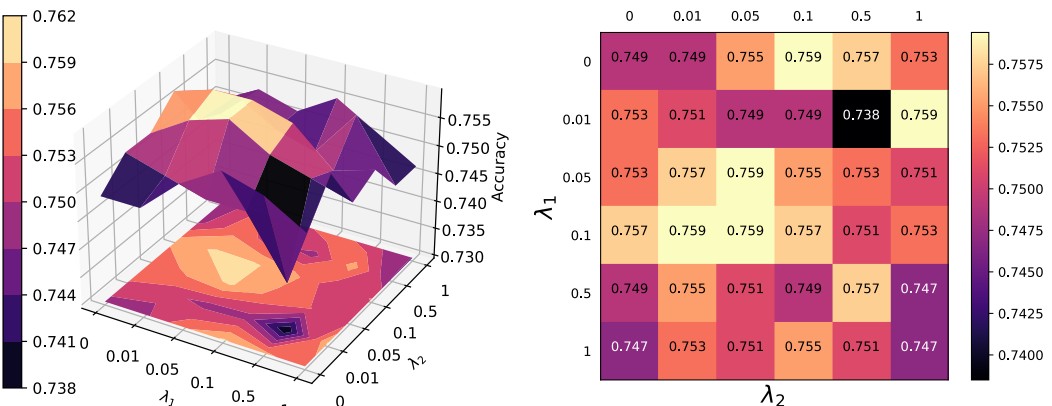

Figure 5: Impact of the weights of regularizers on the recognition rate of KNN on ImageNet10. The 3D contour plot is generated from the discrete heatmaps displayed on the right-hand side.

shown in Table 3, the reconstruction loss is the most significant factor influencing the quality of the learned representations. Using GID and LID loss separately is insufficient for the learning of AE-IDC. Therefore, ID constraints should be combined with the reconstruction loss, and there exists a synergistic relationship between the GID regularizer and LID regularizer.

**Stagewise Training.** In the following, we compare the two-stage training with two training variants: one-stage layerwise training and one-stage global training. Table 4 demonstrates the efficacy of the two-stage training, which is superior to both one-stage layerwise training and one-stage global training. The results in Table 4 also validate that one-stage layerwise training or global training can learn more effective representations, compared to the baseline. In resource-limited situations, using one-stage global training can be an option to reduce the training time.

**Weights of Regularizers.** We investigate the impact of two critical hyper-parameters $\lambda_1$ and $\lambda_2$: the weights for the GID regularizer and the LID regularizer respectively. Fig. 5 demonstrates that IDC is not highly sensitive to the choice of these weights. This alleviates the need for extensive fine-tuning and facilitates the implementation of our approach on the customized dataset.

## 5 CONCLUSIONS

In this work, we present a low dimensional representation learning approach, dubbed AE-IDC, which leverages the global and local ID information of representation. By imposing an ID-aware constraint in the traditional reconstruction loss, we drive the training of AE to preserve the local and global geometric information of representations. Our experimental results depict the superior performance of AE-IDC in solving downstream tasks via efficient and low-dimensional representations. We hope that the findings in this work will motivate the further exploration of more representation learning algorithms from the perspective of regularizations on the dimensionality of data representations.

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
