# A APPENDIX

## A.1 DERIVATIVES OF IDC.

The differential of IDC with respect to the weight $W$ of networks is given by

$$\frac{\partial}{\partial \mathbf{W}} \left( \frac{\text{tr} \left( \mathbf{W} \mathbf{X}^\top \mathbf{X} \mathbf{W}^\top \right)^2}{\text{tr} \left( \mathbf{W} \mathbf{X}^\top \mathbf{X} \mathbf{W}^\top \mathbf{W} \mathbf{X}^\top \mathbf{X} \mathbf{W}^\top \right)} \right) = 4 \left[ t \cdot T_1 - t^2 \cdot T_2 \right],$$

$$\text{where } t = \frac{\text{tr}(\mathbf{C})}{\text{tr} \left( \mathbf{C}^2 \right)}, \ T_1 = \mathbf{W} \mathbf{X}^\top \mathbf{X}, \ T_2 = \left( \mathbf{C} T_1 \right).$$

(3)

## A.2 DETAILS OF EXPERIMENTS.

Table 5: Configuration of Datasets

| Dataset | Training Set | Testing Set | Class |
|---|---|---|---|
| Extended Yale B Georghiades et al. (2001) | 2314 | 1874 | 38 |
| Caltech101 Fei-Fei et al. (2004) | 6907 | 1770 | 101 |
| ImageNet10 Russakovsky et al. (2015) | 13000 | 478 | 10 |
| ImageNet-1K Russakovsky et al. (2015) | 1281167 | 50000 | 1000 |

We describe the CNNs-based AE architectures used in the main text. The architecture of SAE used on Extended Yale B is shown in Fig. 6. Note that the left-hand side of Fig. 6 is the whole architecture of SAE, which comprises four layers. The first two layers encode inputs into embedding representations, and the last two layers decode embedding representations into outputs. The right-hand side of Fig. 6 shows the AE architectures of the first and the third layer of SAE. Note that DAE and SparseAE follow the same architecture as SAE.

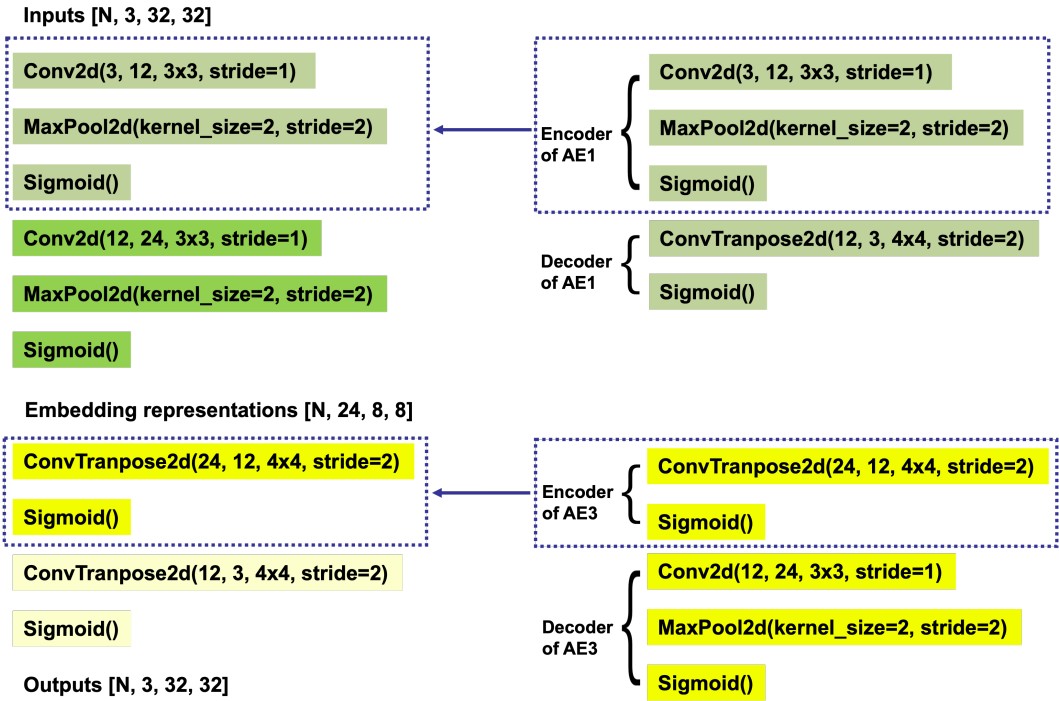

Figure 6: Architecture of the CNNs-based SAE on Extended Yale B.

### A.3 Loss landscape of CNN-based AEs on Extended Yale B.

We present the loss landscape of SAE and SAE-IDC during two-stage training in Fig. 7. Since some loss items have different scales, it is difficult to exhibit the trend of curves having a small scale if we put them in the same subfigure. Therefore, we have depicted them in different subfigures.

Fig. 7 reveals that the GID and LID losses between the input data and its reconstruction data converge to a lower level on both layerwise and global training stages after applying the ID regularizer. Without the ID regularizer, the GID and LID losses of SAE would not converge or converge to a higher level.

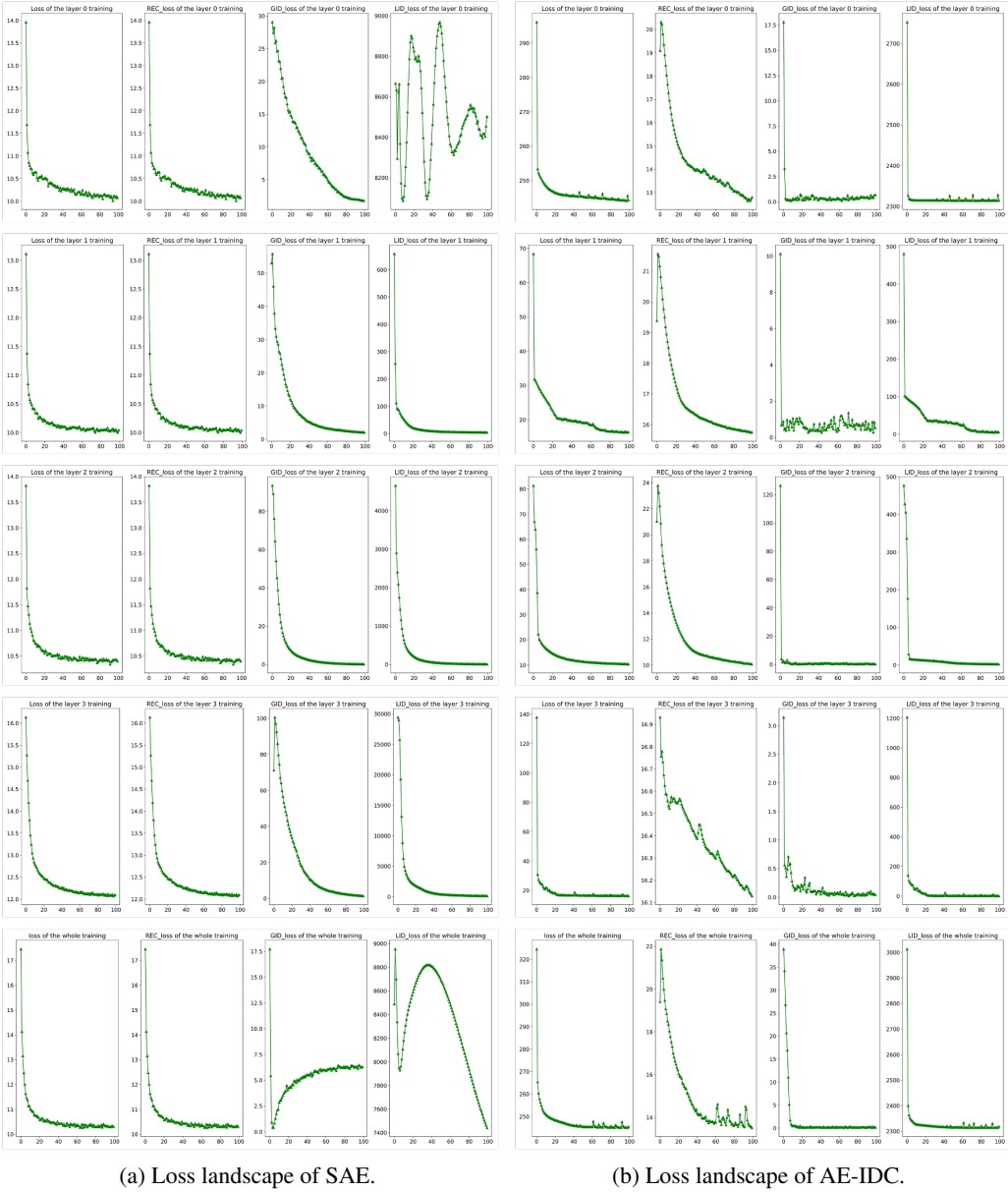

(a) Loss landscape of SAE.          (b) Loss landscape of AE-IDC.

Figure 7: Loss landscape of four layers SAE without and with IDC. Therein, REC_loss refers to reconstruction loss. The first four rows show a loss at the layerwise training stage, and the last row represents a loss at the global training stage. The two models exhibit quite differently, especially at the first AE during the layerwise training stage and the global training stage.

## A.4 VISUALIZATION OF RECONSTRUCTIONS OF DIFFERENT MAEs.

Figure 8: Reconstructions of a 75% masked image from ImageNet through MAE, MAE with GID regularizer, MAE with a LID regularizer and MAE-IDC, from top to bottom.

In Table 6, we present the PSNR and SSIM values for the reconstruction results in Fig. 8. We observe a minor decline in the quality of reconstruction when constraints are applied.

Table 6: Comparison of reconstruction quality.

| Method | MAE | MAE-LID | MAE-GID | MAE-IDC |
|---|---|---|---|---|
| PSNR/SSIM | 22.29/0.74 | 21.98/0.73 | 20.03/0.70 | 20.31/0.70 |