# OpenReview forum: "Autoencoders with Intrinsic Dimension Constraints for Learning Low Dimensional Image Representations"
_ICLR.cc/2024/Conference — ICLR 2024 Conference Withdrawn Submission_

### Official Review · Reviewer_XqLA · 2023-10-18

**Soundness:** 3 good
**Presentation:** 3 good
**Contribution:** 2 fair
**Rating:** 5
**Confidence:** 3

**Summary:**

The authors present a new class of Autoencoders which, in addition to the classical reconstruction objective, aim to preserve topological features such as local and global intrinsic dimensionality. This new autoencoder coined AE-IDC, is trained on a new loss function that penalizes deviations of the local and global intrinsic dimensionality from the reconstructed samples compared to the training data. Experiments on different datasets are conducted to study the effectiveness of the method by using the extracted features on downstream classification and clustering tasks.

**Strengths:**

The paper is well-written. To incorporate local and global features of the data-manifold into the objective function for representation learning is well-motivated, and, to the best of my knowledge, novel. In particular, the paper introduces a new notion of local intrinsic dimensionality of a single image based on the correlations in the channel dimension only. The complexity analysis shows that the proposed penalty terms do not add significant training complexity and can thus be easily integrated into standard autoencoder training for images. Performance gains were observed on different datasets.

**Weaknesses:**

The main goal is to find better low-dimensional representations for images using AutoEncoders. Therefore, the method is righteously tested on the usefulness of low-dimensional latent representations for downstream tasks. However, the proposed new learning objective does not deal with the lower-dimensional representations explicitly. Indeed, both encoder and decoder weights are trained to satisfy the GID and LID constraints. As no theoretical argument is provided for why the proposed penalty terms should make the low-dimensional representations better, it is not clear to me that the proposed objective leads to the desired result. What if only the encoder part is trained on the new objective involving the GID and LID constraints while training the decoder part on the reconstruction loss? This would, in my opinion, compensate for the lack of theoretical evidence and align the algorithm with the main goal of the paper.

The method crucially depends on the intrinsic dimensionality estimator used to calculate the GID and LID. In the paper, the effective number of principal components is used. However, this number was derived in the context of neural activity with no theoretical guarantee. It is not clear why this estimator is suitable for the image domain. Estimating the ID on image manifolds is actually an active area of research, see e.g. [1]. Overall, I would have appreciated a greater discussion on the choice of the ID estimator. Especially, could there actually be a drawback when using a wrong ID estimator? Also, since the ablation studies show the importance of LID, I would have liked to see a greater discussion on this new, somewhat non-standard notion of local intrinsic dimensionality.
In that regard, I think the paper would benefit from a simple toy example where a known manifold, e.g. a swiss-roll, is learned using a standard AE and AE-IDC. Like this, the effect of different ID-estimators on the latent representation can be visualized. If the AE-IDC is able to unroll the swiss-roll providing a more meaningful latent structure than a classical AE, this would strengthen the main message of the paper by confirming the overall intuition. Of course, in that case, only the GID constraint can be used. However, extensions of that toy example to image manifolds are also possible, e.g. by applying the method on image manifolds generated by a StyleGan, see [1].

The experimental evidence provided is not strong in my opinion. I don't see any difference between the quality of learned representations of MAE and MAE-IDC on ImageNet10 (Figure 4 left). Also, the gain in classification performance on ImageNet-1K is not significant (0.1%). The latter suggests that the claim of the authors, namely that "ImageNet10 provides a fast test on ImageNet without loss of generality" is wrong. Thus the remaining results, which do show an advantage of the proposed method, could be due to the low number of classes which weakens the overall message of the paper.

On the basis of this analysis, I am leaning-to-reject the current version of the paper. However, I am happy to change my assessment if the authors can address my raised concerns.

**Questions:**

1. How sensitive is the method to the choice of the intrinsic dimensionality estimator?
2. What happens if we use the method for non-image datasets? Do we see an improved latent space?
3. Do you have any theoretical argument or any intuition as to why the proposed penalty terms lead to better low-dimensional representations?

*Minor questions/comments*:

+ It is curious to consider the channel variability as local intrinsic dimensionality. Can you elaborate on that? It is a somewhat different interpretation of the local intrinsic dimensionality than usual. I see the benefit in not having to compute nearest neighbors, but still find the use of terminology confusing.
+ I don't quite understand Figure 3. The sub-titles describe the loss functions used. However, in each plot, three loss functions are depicted. In the sub-plot with the title "Loss landscape with local ID constraint" I would have not expected to see the plot referring to "gid_loss".
+ It seems there is an issue with the referecing. Figures, algorithms and chapters have 2 red boxes in my version.
+ Section 3, notation: Y is not used. W is referred to transformation matrix and later in the text as weight parameters. I don't see where the definition of Y is used and needed.
+ also, there is no real need to introduce the covariance matrix
+ Table 1: Why are results on ImageNet-1K not bold but only underlined?
+ Page 8 "Results on Classification Tasks.": you write that improvement range is 1% - 5 %. However, I see a maximum of improvement of 4.22 %. Why not being precise here?
+ You write: "For a fair comparison, we let the compared methods use its public official fine-tuned models without any modification. as shown in Table 1. Though only improving the best performance by 0.1% on ImageNet-1K, the training process of MAE-IDC is more concise and easy to understand." Please elaborate. Does that mean you don't do any fine-tuning for MAE-IDC?

---

### Official Review · Reviewer_dwAb · 2023-10-31

**Soundness:** 3 good
**Presentation:** 2 fair
**Contribution:** 1 poor
**Rating:** 3
**Confidence:** 4

**Summary:**

This work proposes a regularizer for generative models based on measures of intrinsic dimensionality (ID) and carries out an empirical study of its merits. The measure of ID used is based on squared sums of normalized eigenvalues of covariance matrices of sample batches. Two variants of this ID measure are considered, which they call global and local, where the former involves matrices of shape "batch size" x "product of channel, height and width" and the later involves matrices of shape "channels" x "product of height and width". A stacked-style autoencoder is proposed where hidden layers in the first half and smaller than the input dimension ("undercomplete"), and the hidden layers in the second half are greater than the input dimension ("overcomplete"). A sequential layer-wise training scheme is proposed using the proposed regularizer. The quality of the obtained representations are evaluated on downstream classification and clustering tasks on small image benchmarks and subsets of imagenet. The benefits of the regularizer are compared in several autoencoder flavors, and improvements on the order of 1-5% are claimed using the regularizer. Some experiments on fine-tuning vision transformer models (ViT) with the proposed regularizer are made.

**Strengths:**

Using geometric properties of data to improve generative models is an interesting problem. The authors make a serious attempt to study one aspect of this problem, and I appreciate their enthusiasm. The empirical study on the models they chose is relatively comprehensive.

**Weaknesses:**

I have some doubts on the relevance of this work to the ICLR community. Autoencoders, although historically important, are less relevant to today's generative modeling community due to the rise of high-capacity models such as diffusion models. It is sometimes okay for a work to focus on autoencoders, e.g. [0], especially for new mathematically sophisticated techniques for which theoretical analysis that is not yet possible on other kinds of models. It would be more relevant if the authors considered diffusion models. However I acknowledge that perhaps this work is a step towards using the regularizer in more advanced models.

The relationship of the proposed ID measure to other ID measures in the literature is not explored. For instance, does the ID measure in this work correlate with the other measures discussed in Section 2.2 on toy-benchmarks (e.g. hyper-spheres/cubes) or real-images?

One nitpick: in Section 2.3 you write that [1] showed "the dataset with higher ID value is more vulnerable to attacks", but actually that work does not consider adversarial attacks at all.

I am also not very clear on why the proposed local ID measure is actually local. In Section 3.1, the authors write "[c]onsidering that feature maps from the same data point are highly correlated", but I don't see why this is the case.

Regarding the cost of the regularizer, I read asymptotic complexity analysis in Section 3.3, but I still have some doubts it is not more expensive. Wall-clock timings for the regularized and unregularized would be helpful.

Another nitpick: the term "loss landscape" usually refers to the function L(w) for different weights w (often projected into a low-dim space). The authors use that term to describe the curves  in the Figure 3. These are usually just called "loss curves".

Plots in Figure 4 are hard to read.

[0] Topological Autoencoders - Moor et al. (ICML 2020)
https://proceedings.mlr.press/v119/moor20a.html

[1]The Intrinsic Dimension of Images and Its Impact on Learning  - Pope et al. (ICLR 2021)
https://openreview.net/forum?id=XJk19XzGq2J

**Questions:**

Can the authors show their regularizer can improve modern generative models, e.g. diffusion models?

Can the authors justify their claim that their "local" ID measure is actually local?

Can the authors provide timing info for training with the regularizer versus without it?

What is the relationship between their ID measure and other ID measures? E.g. the MLE estimator or variants of Levina and Bickel

---

### Official Review · Reviewer_UAX2 · 2023-11-05

**Soundness:** 3 good
**Presentation:** 3 good
**Contribution:** 3 good
**Rating:** 6
**Confidence:** 4

**Summary:**

In this paper, the authors observe that existing autoencoder models using distance-based reconstruction loss may be missing important local characteristics of the data. Instead, they argue that both local and global intrinsic dimensionality information is needed in order to assure the fidelity of reconstruction to the original inputs. The main contribution of the paper is the proposal of a differentiable regularization that penalizes two measures of the dimensional discrepancy between the input and its reconstruction: the squared difference of global ID (GID), and the average squared difference of local ID (LID). To calculate GID values, they make use of an estimator due to Rajan et al. based on normalized eigenvalues of the correlation matrices, and to calculate LID values, they apply this same estimator over local feature maps. In their experimentation, the authors implement their approach on several image datasets for classification and clustering tasks using several autoencoder variants applied to CNN and vision transformer architectures.

**Strengths:**

S1) Although the authors themselves do not point this out, ID-aware regulation of reconstruction could be important due to the interplay between local distance distribution and dimensionality: there is a known tendency for distances within localities of low ID to have significantly higher variation than when the locality has high ID, and this is not taken into account by standard autoencoder loss functions such as MSE.

S2) The experimental results provided for the classification and clustering tasks do show a substantial and consistent improvement in accuracy (for classification) and ARI/AMI (for clustering) for their ID-based autoencoder regularization, as compared to their unregularized counterparts. These include ablation studies that show that performance is not overly sensitive to the choice of the regularization parameter values.

S3) The paper presentation is of a generally high standard: well-organized, well-written, and clear.

**Weaknesses:**

W1) Among the technical contributions of the paper, the only novel idea is a rather straightforward regularization based on differences of IDs (global and local) between the inputs and their reconstructions. The relationship mentioned in S1 between local distance distributions and LID may not be properly accounted for or fully exploited, and so there is likely more work that needs to be done here. (q.v. the LID model used in the two papers by Ma et al.)

W2) Although the authors make the case that ID regularization does lead to improvement in autoencoder-based applications, it is not clear whether they are sufficient to allow them to compete with state-of-the-art contrastive learning models. It would have been good to see this question settled with a more extensive experimentation.

**Questions:**

Please address W1 and W2.

---

### Official Review · Reviewer_3713 · 2023-11-07

**Soundness:** 2 fair
**Presentation:** 3 good
**Contribution:** 2 fair
**Rating:** 5
**Confidence:** 4

**Summary:**

The paper proposes a geometry-aware representation learning method based on the intrinsic dimensionality of data. It penalizes reconstruction losses in autoencoder-inspired models to preserve global and local geometric structures.

**Strengths:**

The paper is clearly written with good organization. The problem dealt with is crucial and the article does a good job at building up to the problem statement with ample discussion on related works.

**Weaknesses:**

The model proposed seems incremental based on regularization over well-established architectures. There are novelty concerns regarding the training process as well. Qualitative results to empirical findings also seem inadequate.

**Questions:**

1. The paper begins with a definition of ID from a model selection perspective "*ID for a set of data points is the minimum number ... little information loss*". Is there a definitive notion to this (e.g. the Minkowski dimension or the upper Wasserstein dimension), since in modern empirical process theory used in machine learning, it is often considered that the distribution modeling the data is infinite-dimensional. In other words, there exists no finite-dimensional parametric space describing the distribution fully.

"*Most AEs involve dimensionality compression first ... ID plays a crucial role.*" This needs further elaboration. Is the latent dimension (the space reached after "dimensionality compression") somehow linked with the ID?

2. If ID is the minimum number of directions required to explain the underlying distribution, how does it differ globally and locally?
Can you mathematically relate global and local ID, since Section 2.2 hints at it?
Also, looking at the definitions of LID and GID; is the former an aggregate over the values of the latter from batches? If that is true, do we need to include both as regularizers, as it runs the risk of over-penalization?

3. What is the motivation behind selecting $L_{2}$ loss to reconstruct, as its use is impractical in case of high-dimensional data, e.g. image?

4. [Section 3.2] What is meant by under/overcomplete? How do you decide on a suitable model architecture given a dataset? The discussion on this [Appendix A.2] seems inadequate.
Why follow the SAE architecture in the first place?

5. Is there any explanation as to why penalizing with only local ID constraint makes the GID loss explode? [Figure 3]

---

### Official Review · Reviewer_mtEJ · 2023-11-07

**Soundness:** 2 fair
**Presentation:** 2 fair
**Contribution:** 1 poor
**Rating:** 3
**Confidence:** 4

**Summary:**

In this paper, the authors attempt to design an extension of Stacked AutoEncoder (SAE) by incorporating the global and local Intrinsic Dimension (ID). The whole training framework is based on the existing SAE.

**Strengths:**

- The paper is easy to follow.
- The idea of incorporating GID and LID is testified on MAE.

**Weaknesses:**

- The novelty seems incremental. The motivation of AE-IDC lacks insights. Specifically,
    - The Intrinsic dimension is not a new concept. It lacks insights to incorporate ID into AE/SAE by simply adding minimizing $ID(X) - ID(\hat X)$. There are plenty of works adopting diverse terms for AE.
    - The training paradigm is almost the same as the classical SAE[1]. It is not new.

​	Overall, the contributions may not be enough for ICLR.

- The improvements shown in experiments seem incremental. The improvement is foreseeable when some regularizations are incorporated into AE/SAE.
- The authors fail to clarify why ID is suitable for image data.

[1] Stacked denoising autoencoders: Learning useful representations in a deep network with a local denoising criterion,  JMLR, 2010.

**Questions:**

- What does **M** in LID mean? The formal definition is missing.
- Since AE and SAE do not rely on the type of data, can AE-IDC work on non-vision data? Why do the author highlight the image representations?